# OpenReview forum: "LION-DG: Layer-Informed Initialization with Deep Gradient Protocols for Accelerated Neural Network Training"
_ICLR.cc/2026/Workshop/Sci4DL — Submitted to Sci4DL 2026_

### Official Review · Reviewer_u4xo · 2026-02-26

**Fit:** 2
**Significance:** 2
**Confidence:** 3

**Summary:**

The paper proposes a modification to the initialization strategy for "deeply supervised" image classification convolutional architectures, namely, initializing the auxiliary classification heads to 0. The idea is to reduce their interference with the optimization of the backbone at early times.

**Strengths:**

- The main idea is simple and easy to implement. It is hyperparameter-free and has no extra costs.

**Suggestions:**

Overall, the writing is clean and concise, but the experiments need to be designed more carefully to support the claims more strongly.

- In general, performance comparisons are only convincing if results with extensive hyperparameter tuning are shown. For example, the 8-10% speedup to reach a certain accuracy can easily change if hyperparemeters like learning rate, weight decay, lr-schedule etc. are changed.
- Gradient ratio for He init vs LION-DG (figure 1 right) seems to fluctuate and cross over several times. Step 45 in table 2 showing the difference in gradient-ratios seems cherry-picked to assert Proposition 2. Maybe this can be addressed by further averaging or designing a more robust measure.
- As already mentioned in the limitations, the method not working with SGD+momentum is a major drawback since it is quite common to use SGD for image classification. The method would be stronger if it can be modified to be optimizer-agnostic.
- There is a subtle mismatch between what Proposition 2 claims and what the empirical results show. The empirical results are meant to support controlled gradient growth on the backbone parameters (coming from auxiliary objective), while Proposition 2 claims linear growth. This should be made more precise.

---

### Meta-Review · Area_Chair_9rxo · 2026-02-28

**Recommendation:** Reject

**Metareview:**

This paper proposes to zero-initialize auxiliary classification heads in so-called "deeply supervised" classification networks. First, the fit to the workshop is tenuous as best, as the main goal of the paper is to improve performance rather than further our understanding of deep learning. Second, as noted by the reviewer, the experiments to validate the method are not carefully controlled and may be deceiving: effects are tiny and noisy. Third, several of the measured quantities are not defined in the text ("gradient ratio", "cosine similarity"). I therefore recommend rejection.

---

### Decision · Program_Chairs · 2026-03-02

Reject